# OpenReview forum: "ChatBug: A Common Vulnerability of Aligned LLMs Induced by Chat Templates"
_NeurIPS.cc/2024/Workshop/SafeGenAi — SafeGenAi Poster_

### Official Review · Reviewer_EFXV · 2024-10-10
**Reviews for ChatBug Vulnerability in Aligned LLMs**

**Rating:** 7
**Confidence:** 3

**Review:**

Overall: This paper provides valuable insights into vulnerabilities in aligned LLMs for chat templates. The specific attacks within the two studied attack frameworks have been explored in existing research and some competitions. However, this study provided comprehensive experiments and critical insights and discussions that can inform future countermeasures.

Pros:
1. Important topic: It deals with important issues related to the safety of LLMs with general-purpose capabilities and a large amount of knowledge.
2. Comprehensive insights and experiments: The insights into the vulnerabilities described in this study are comprehensive, and many experiments have been conducted to verify them.
3. Discussion of future measures: Approaches to mitigating the vulnerabilities identified by the study are discussed.

Cons:
1. Countermeasure experiments: We d a workable solution to ChatBugs that optimizes the tradeoff between usability and effectiveness.

---

### Official Review · Reviewer_AqGs · 2024-10-10
**ChatBug: A Common Vulnerability of Aligned LLMs Induced by Chat Templates**

**Rating:** 6
**Confidence:** 5

**Review:**

Strengths:

- The exploration of safety risks introduced by chat templates is quite interesting. Message Overflow Attacks and Format Mismatch Attacks, as extensions of jailbreak techniques, highlight the potential risks involved in instruction tuning for large language models (LLMs).
- The work offers valuable insights into how malicious users can exploit vulnerabilities (i.e., ChatBug) within predefined chat templates to bypass safety alignment mechanisms, an important topic in ensuring safe deployment of LLMs.

Weaknesses:

- The experiments could be more comprehensive by including LLMs from other languages, such as ChatGLM or Qwen, to further validate the generality of the conclusions.
- It would strengthen the evaluation if GPT-4o were used as part of the assessment.
- It's unclear whether the proposed method can handle multi-turn jailbreak scenarios. Further discussion or experimentation on this aspect would be beneficial.